# A Narrative Inquiry into the Adjustment Experiences of Male Bladder Cancer Survivors with a Neobladder

**DOI:** 10.3390/ijerph17218260

**Published:** 2020-11-09

**Authors:** So Hee Kim, Eunjung Ryu, Eun-Ju Kim

**Affiliations:** 1Graduate School, Chung-Ang University, Seoul 06974, Korea; ljoy@naver.com; 2Department of Nursing, Chung-Ang University, 84 Heuksuk-ro Dongjak-gu, Seoul 06974, Korea; 3Department of English, Hanyang Women’s University, Seoul 04763, Korea; exk188@gmail.com

**Keywords:** urinary bladder, neoplasms, adjustment, cystectomy, survivors, male

## Abstract

This study aimed to explore three male bladder cancer survivors’ adjustment experiences after neobladder reconstruction. A narrative inquiry method was adopted to closely investigate the individual experiences of bladder cancer survivors and the meaning of their experiences. Three themes emerged regarding physical and mental changes resulting from neobladder reconstruction: difficulty urinating or holding urine, sexual dysfunction and sexlessness, and stress resulting from urinary incontinence. Life changes following surgery varied across each participant and included ‘unwanted retirement’, ‘quitting drinking and leaving work’, and ‘beginning of a restrained life’. The theme of adjustment experience emerged, comprising ‘active exploration of resolutions’, ‘accepting change and partial return to daily life’, and ‘living in line with the health condition and family wishes’. Bladder cancer survivors with a neobladder, in this study, continue to adjust to changes in the voiding mechanism, various symptoms including incontinence, and life changes even after surgical cancer resection. The findings suggest that not only therapeutic interventions, but additional interventions are also needed to assist bladder cancer survivors with adjustment, rehabilitation, and return to society. These findings are also expected to be used both to educate bladder cancer survivors with a neobladder and to develop policies to help them.

## 1. Introduction

The standard treatment for muscle invasive bladder cancer (MIBC) is radical cystectomy followed by urinary diversions, such as ileal conduit or orthotopic ileal neobladder [1]. While ileal conduit requires a stoma and urine bag, a neobladder enables normal urination, because it is created using part of the ileum, which is then connected to the ureter and urethra [2].

A neobladder may improve quality of life because it enables normal urination after surgery. However, at the same time, patients are also required to learn skills for postoperative self-care, adjust to damaged physical functions, and cope with the postoperative changes [3]. Postoperatively, patients experience incontinence from unrecovered urethral sphincter functions [4]. Moreover, because the functional bladder volume is only about 120 mL immediately following surgery, patients need to urinate every two hours [5]. The patients also need to train themselves to relax the pelvic floor muscles and increase abdominal pressure in a seated posture in order to maintain minimal residual urine [5]. However, while learning and practicing self-catheterisation in cases when they are unable to self-void, patients may experience spasms, pain, and difficulties with work life [3]. In addition, a large number of male survivors lose erectile function after surgery [3]. Even though survivors explore other means to promote their sex life, such as medications, many of them have found the measure unsatisfactory [3].

Fitch et al. [6] reported that bladder cancer survivors who underwent radical cystectomy experienced changes in body function, body image, sexual function, and relationships after surgery. According to Beitz and Zuzelo [3], cancer diagnosis and detesting the idea of a bag were the main issues at the early stage of experience. The mid-stage issues included reacting to postoperative pain and physical changes such as managing tubes and drains, responding to incontinence, self-catheterising, neobladder training and urination with the neobladder [3]. The issues at the later stage were adapting to new toileting needs, keeping the neobladder healthy, adjusting to altered bowel movements, changing sexuality, considering the impact of smoking, seeing cancer as a lifelong threat and reflecting on the neobladder choice [3]. As seen in the studies mentioned here, bladder cancer survivors with a neobladder undergo an array of postoperative changes, and they strive to accept the changes, find strategies to cope with them and lead their lives as best as they can.

However, the main foci of the previous studies were limited to the postoperative problems experienced by survivors [3,6,7]. Consequently, in-depth information in regard to individual patients’ coping and adjusting experiences has been missed. In order to bridge the gap, a qualitative approach is needed, given that it can provide an in-depth understanding of how bladder cancer survivors cope and adjust to the changes after neobladder reconstruction. Further, as sociocultural factors can have a vital impact on cancer survivors’ adjustment [8], the lived experiences of Korean bladder cancer survivors with a neobladder are worth exploring. Lastly, despite the fact that bladder cancer is four times more common in men [9] and that neobladder reconstruction is more frequently performed on men [5], to the best knowledge of the authors, no studies have examined the case of Korean male bladder cancer survivors with a neobladder. Therefore, the findings from this study are expected to provide valuable information for practitioners and patients themselves, in addition to researchers in related areas.

This narrative inquiry aimed to understand the changes male bladder cancer survivors experience after neobladder reconstruction and their coping experience. The research questions were as follows: What is the postoperative change male bladder cancer survivors with a neobladder experience? How do male bladder cancer survivors with a neobladder cope with and adjust to the postoperative changes? What are the meanings of the experiences male bladder cancer survivors have with a neobladder?

## 2. Methods

### 2.1. Study Design

This study adopted the narrative inquiry proposed by Clandinin and Connelly [10]. The interest of narrative inquiry is not in the quantity but in the quality of data. That is, in narrative inquiry, one participant is enough when the participant’s story is interesting enough to be explored [11]. Through the participants’ narratives, the study qualitatively explored three Korean male bladder cancer survivors’ experiences in regard to the changes they encountered following surgery, coping process and adjustment.

Complying with the tenets of a narrative inquiry [12], the authors not only looked into the individual stories of bladder cancer survivors with a neobladder and their underlying meanings but also the social, cultural and institutional narratives that constituted, shaped and expressed their experiences.

### 2.2. Study Participants

Three Korean male bladder cancer survivors with a neobladder were the participants of the study. Since the study was interested in the experience of male bladder cancer survivors with a neobladder, homogeneous purposeful sampling was adopted. According to a prior finding, the reduced quality of life score after surgery increases until 12 months after surgery and is maintained at a stable level thereafter [12] and this implies that the adjustment to major postoperative changes is considered to occur over a period of one year. Thus, the first major criterion for sampling for this study were the patients who had surgery at least one year before study participation to ensure that they had adequately experienced postoperative changes and the subsequent adjustment processes. Second, the study was interested in recruiting Korean male adult bladder cancer survivors because their stories were expected to reveal the social cultural influence on the experience of adjustment among a certain group of patients. This was particularly so given that gender roles have been settled for a long time based on Confucian tradition in Korea [13].

The author distributed the recruitment flier to patients who visited the outpatient clinic of two tertiary hospitals in Seoul and explained the rationale and objective of the study. Initially, four male cancer survivors provided written informed consent to participate, but one withdrew consent because of privacy concerns. Thus, three males, aged 54–59 years old, became the final participants of the study. Their postoperative periods ranged from one year, nine months to five years, ten months.

Kim was 54-years old and underwent neobladder reconstruction because of MIBC five years and ten months before the study. He had worked in a global firm for more than 20 years but had to quit his job after being diagnosed with cancer and undergoing surgery. Lee was a 59-year-old taxi driver. He underwent neobladder reconstruction because of MIBC two years and five months before the study. Two years after surgery, he underwent urethrectomy because of cancer relapse and had a suprapubic catheter placed. Park was a 58-year-old man. He underwent trans urethral resection of a bladder tumour (TURBT) for non-muscle invasive bladder cancer (NMIBC) six years before the study. He had to undergo 11 more rounds of TURBT because of continued cancer relapse and progression. One year and nine months before this study, he underwent cystectomy and neobladder reconstruction because of MIBC. All three participants were married.

### 2.3. The Process of a Narrative Inquiry

#### 2.3.1. Being in the Field: Walking into the Midst of Stories

The first author became interested in bladder cancer survivors with a neobladder while providing hospital nursing care to patients who underwent neobladder reconstruction. The author witnessed patients who underwent a neobladder reconstruction experience complications after surgery and have difficulties as they engaged in urination training. The author thus came to think of ways to help the patients and concluded that understanding the postoperative changes and patients’ adjustment to such changes could be a way to begin.

#### 2.3.2. From Field to Field Texts: Being in A Place of Stories

A narrative inquiry has two possible starting points. One begins with listening to individuals’ stories, and the other begins with living alongside participants as they live and tell their stories [14]. This study started narrative inquiry by listening to participants’ stories, that is, through interviews, to examine the overall experiences of bladder cancer survivors with a neobladder during the extended period between surgery and adjustment.

#### 2.3.3. Composing Field Texts

In this study, field texts were composed through interviews conducted by the first author between January 2019 and May 2020, which took place in a quiet park or coffee shop at the participant’s desired time and region. Two interviews were conducted with each participant, with each session lasting 30–120 min. Considering that postoperative changes such as incontinence may be an uneasy topic to disclose to others, the author asked the participants to start their stories from the diagnosis of bladder cancer and proceed chronologically to the decision to undergo urinary diversion, the surgery experience, changes after surgery, coping and adjustment. However, the interview was not bound by the content or order of the questions; the conversation was allowed to flow according to the stories told by the participants. The interviews were audio-recorded, and the author repeatedly listened to the recordings to transcribe the contents immediately after the interviews.

#### 2.3.4. From Field Text to Research Texts: Making Meaning of Experience

The authors repeatedly read the field texts and narratively coded the characters, actions and events, then linked stories that appeared in the field texts [10]. Further, the authors composed interim research texts by paying close attention to the three-dimensional narrative inquiry space—composed of temporality, sociality and place—and identifying the themes, tensions, narrative threads and patterns [10]. A second interview was conducted based on the interim research text, which was followed by another round of composing field texts, repeatedly reading them and composing interim research texts.

#### 2.3.5. Composing Research Texts

The authors strived to represent participants’ and authors’ voices in balance, reflect the diverse voices of participants and reflect both the authors’ and participants’ viewpoints in the research texts. Further, the authors composed the research texts while being aware of the primary and secondary audience. The primary audience were cancer survivors with a neobladder, future patients and their families, whereas the health professionals who provide care for bladder cancer survivors with a neobladder and members of society who will live alongside these survivors were the secondary audience.

### 2.4. Ethical Considerations

This study was approved by the institutional review board of C university (1041078-201901-HR-005-01) prior to beginning the study. Prior to the interviews, the authors obtained informed consent from the participants. The first author explained to the participants the reason for the study, aims, methods, guarantee of anonymity and confidentiality, audio-recording of the interviews and freedom to withdraw from the study without penalty. Relational ethics and responsibility were considered in all steps of the research. The interviewer practiced empathetic listening, suspending critique and mistrust. The interim research texts written after the interview were shared with participants to allow them to say more, clarify, add or delete specific content in the texts. Pseudonyms were used to protect the participants.

## 3. Results

### 3.1. Physical and Mental Changes Following Neobladder Reconstruction and Subsequent Adjustment

Following neobladder reconstruction, the participants experienced some common physical and mental changes, including lower urinary tract symptoms, sexual dysfunction and stress.

#### 3.1.1. Difficulty Urinating or Holding Urine

The greatest change experienced by the participants after surgery involved urination. In contrast to the past when they urinated easily, they now had to urinate by straining or pressing on the abdomen. As it was difficult to press on the abdomen with one hand while standing, they had to sit on the toilet to exert force on the abdomen. Inability to fully empty the bladder was also a problem. Given the small size of the neobladder immediately after surgery and the presence of residual urine, most participants experienced frequent urination and nocturia.
It’s so difficult to void. I can’t void even if I push hard. So even after voiding, I don’t feel as fully emptied as healthy people do. I think I’m not emptying enough. You can press on your lower abdomen only so much, it doesn’t work. Then I just press two or three times and just give up and leave the toilet. I’d think there’s still some left in there. I feel unsettled.(Interview 2, Lee)

It was not only difficult to urinate but also to hold in urine; all participants experienced incontinence to some degree, which varied from heavy leaking to dribbling. The symptoms were worse at night.
Incontinence was the most serious, and that was my greatest concern. At first, it just flows out without any restraint. Later, it’s not like incontinence but I can feel the urge. I feel the urge, but I can’t hold it in”.(Interview 2, Kim)

#### 3.1.2. Sexual Dysfunction and Sexlessness

In addition to various lower urinary tract symptoms, changes in sexual function were common. All participants had marked sexual dysfunction following surgery, and an altered body image caused by incontinence affected their ability to resume their normal sex life.
My sexual functions are dead. I can’t get an erection. I asked the doctor about regaining the function, but I was told there is only about a 30% possibility.(Interview 2, Kim)

#### 3.1.3. Stress from Urinary Incontinence

The participants experienced stress from incontinence. While incontinence did improve over time, inability to have complete control produced anxiety and frustration.
With incontinence, I feel wet and heavy with diaper because of full absorption. Then the overflowed volume can leak if I don’t change it quickly enough. The heating sensation from incontinence causes quite a bit of discomfort as well. It truly ruins the quality of life if I don’t keep changing diaper on time.(Interview 1, Park)

#### 3.1.4. Coping with Physical and Mental Changes

The participants learned to adjust to and cope with incontinence and the changed their voiding mechanism. The most common and basic coping strategy was to wear a diaper and do Kegel exercises, which effectively improved incontinence. As a result of these actions, some participants experienced improvements in incontinence over time, varying from three months to one year. Daytime incontinence improved first. The severity of night-time incontinence varied widely, from very mild to severe, and unlike daytime incontinence, it was not completely resolved.
I wore diapers. Whatever I do, whether I take the subway or what else, I wore diapers day and night and packed these many extra diapers in my bag. They tell me to do sphincter exercises, and I did. Frequency and volume of incontinence were reduced.(Interview 1, Kim)

Two participants also performed self-catheterisation for urinary frequency due to residual urine and unimproved incontinence. Although they were resistant to self-catheterisation and had discomfort at first, once they performed it and drained the neobladder, they were able to remain free from frequent urination or incontinence for a few hours. In addition, the discomfort with self-catheterisation subsided once they became familiar with the procedure.
It was hard at first. But later you get better at it. You feel a little stiff when you insert the catheter, but you get to be so much more comfortable once you do it.(Interview 1, Lee)

Although the participants developed sexual dysfunction and could no longer enjoy their previous sex life, they demonstrated an accepting attitude. One participant reported an active sex life before surgery while others were already sexually inactive before surgery. Irrespective of their pre-surgery level of sexual activity, the inability to enjoy sex after surgery was frustrating and regrettable. However, they accepted it because it is an irreversible change.
I eventually gave up. You can’t get it back. It’s irreversible. You can’t have it back.(Interview 1, Kim)

Regarding stress from incontinence, the participants accepted their present selves, knowing that they would not recover to their preoperative condition. Instead of focusing on imperfect functions, they gradually adjusted to the change and accepted themselves as people not different from others.
I used to be kind of sensitive, but I’ve become insensitive, adjusted and controlled myself.(Interview 1, Park)

### 3.2. Changes in Life Resulting from Neobladder Reconstruction and Subsequent Adjustment Experiences

Changes in life after surgery varied across participants, and adjustment experiences differed according to the respective changes.

#### 3.2.1. Kim’s Story

Changes in daily life: Unwanted retirement

Kim was 54-years old and underwent neobladder reconstruction because of MIBC. The most difficult time after surgery was the first year, because during this time the lower urinary tract symptoms were the worst, and he lost his job. When he returned to his job six months after surgery and chemotherapy, his work was reassigned to someone else. He was not offered another position and eventually decided to retire voluntarily. After resigning, finding another job in the same industry was challenging.
(After my leave of absence) When I returned, my work was gone. At first, my boss said he’ll give me another position but that didn’t work out… I just passed time. I just surfed the internet in the office. It was a tough time for me. You know, it’s unsettling that you don’t have any work to do. The company was taking applications for voluntary retirement, so I just submitted my application in the first round.(Interview 2, Kim)

Adjustment experiences: Active exploration of resolutions

Kim accepted the changes that occurred after surgery but actively sought solutions and coped with the situation, seeking reemployment education and attending job interviews. He had incontinence, but he wore a diaper and resumed being active, striving to alleviate the symptoms by doing Kegel exercises and fabricating a penis clip on his own to manage incontinence. As a result, his incontinence frequency and volume began to decrease six months after surgery, and one year after surgery it had decreased to the point that he no longer needed a diaper or pad. Further, he was offered a new job at a foreign company in the same industry, and he was assigned to a position that involved overall planning and running projects, based on his past employment experience.
I searched the internet (to do Kegel exercise) (omitted). There’s this thing that holds in incontinence. It gives pressure to prevent leakage. But it’s not efficient because there’s only a few sizes. So, I made one for myself. I drew the design and told a friend of mine in the plastic industry to make it for me.(Interview 2, Kim)

Meaning of the adjustment experiences: Return to a daily life

Kim had been busy with work since graduating from college, and his wife was also busy with work. His son was studying abroad, so his family never had a chance to live together. As a result, he naturally structured his life around his work, which was very important to him. The bladder cancer diagnosis and surgery took away the most important thing to him, his work. Adjustment meant returning to the life he had before surgery, that is, his work. The support of his co-workers was a great motivator for him to return to work. Further, Kim was relatively young at the time of surgery, in his late 40s, and he had a lengthy professional career, which helped him eventually return to work. Kim believes that the bladder cancer diagnosis and surgery gave him the opportunity to work abroad and he is satisfied with his current life.
During 26 years of working, I seriously invested all my time in work for nearly 10, 15 years. I didn’t even go to college alumni meetings. I naturally became work-centred. I lived my life wrong. I have nothing to talk about other than work.(Interview 2, Kim)

Returned crisis and coping: Re-surgery as a result of unhealthy behaviours and compromise

One year after surgery, Kim was able to return to the life he had before surgery. He naturally began to drink again as he worked and socialised with his co-workers and the drinking gradually increased in quantity. This eventually led to a ruptured bladder from excessive drinking five years after surgery, which required a re-surgery. Although he was aware that smoking is associated with bladder cancer, and had needed re-surgery because of excessive drinking, he continued to smoke and drink. He had several failed attempts to quit smoking prior to the bladder cancer diagnosis and believed that stress was a greater problem than smoking. He was resistant to his family’s concerns about his smoking because he felt they are treating him like a patient, which he found unreasonable when he was healthy again after surgery and not in need of any treatment related to bladder cancer. Thus, Kim tried to justify that the second surgery was not a big problem because it was not due to a cancer relapse, and that, although smoking is a cause of bladder cancer, stress may also be a cause.
‘Don’t drink’, ‘don’t smoke.’ Everybody totally treats me like a patient. My wife does; my family do; my younger siblings do, even my parents treat me like a patient.(Interview 1, Kim)

However, after the re-surgery, Kim began gradually changing his lifestyle, reducing his consumption of strong alcohol and changing to e-cigarettes out of concern that he would develop a health problem in a foreign country.

#### 3.2.2. Lee’s Story

Changes in daily life: Quitting drinking and leaving work

Lee was a 59-year-old taxi driver. He underwent neobladder reconstruction because of MIBC two years and five months before the study. He had residual urine and incontinence after surgery but did not experience many life changes. He initially stopped working, then gradually returned to work and quit drinking.

Adjustment experiences: Accepting change and partial return to daily life

Lee positively coped with the physical changes that occurred after surgery. Although he had incontinence, he wore a diaper and engaged in exercise he had enjoyed before surgery, and he continued to meet people. After doing Kegel exercise every chance he had, his incontinence subsided three months after surgery, and he was able to live diaper-free in the daytime. Although incontinence did not stop completely, he willingly accepted the symptom as one that can occur in anyone with ageing, even without bladder cancer. Further, when he faced difficulties in his daily life associated with residual urine and urinary frequency, he performed self-catheterisation that he learned in the hospital, which gave him 3–4 h to work and earn a living.
You have to accept what you must accept and overcome it. You could have a little incontinence, so what? Everyone develops incontinence when they’re about 70. Who wouldn’t have incontinence when they’re 100?(Interview 2, Lee)

Meaning of adjustment experience: New normal—partial returning to daily life and changing into a family man.

Lee had several lower urinary tract symptoms but still partially maintained the life he had before the surgery. He informed his close friends about his difficulties, maintained social relationships, engaged in activities and gradually resumed work, which was possible because private taxi driving allowed him to work a flexible schedule and freely return home for self-catheterisation when needed, then quickly return to work.
I talk to people about my illness. If you’re all crawled up, you can’t overcome your illness. You have to actively move around. I still exercise outside, play soccer even in my diapers. I don’t drink but I hang out with others when they’re drinking. I eat rice, meat, and drink soda instead. Otherwise you can’t overcome your illness.(Interview 1, Lee)

Although he could not continue with his sex life following the surgery, his relationship with his spouse actually improved. He had no difficulties in his marital relationship as a result of the changes in sexual activity after surgery because he attended preoperative counselling with his spouse and consented to surgery even after being informed about the postoperative changes in sexual functioning.
Well, sex is now out of the question because even that part that ejaculates the semen was removed. That I can’t even begin.(Interview 1, Lee)

In the past, Lee had run a business that supplied parts to a large corporation, so he had to wine and dine with his clients every day, which made him neglect his family. He drank with his friends every day and continued to put himself at the centre of his life even after he began driving a private taxi. However, after the surgery, he realised the value of family and changed his life to centre around his family, taking family trips and helping with household chores, which strengthened his relationship with his spouse and other family members.
Being sick, I learned that my wife is the only one who would take care of me and my kids are the only ones who care about me. Then I realized what family really is. After that, I did a one eighty. My kids like it and my wife loves it. My kids give me strength and motivate me to live.(Interview 2, Lee)

Returned Crisis and Readjustment: Changes due to Relapse and Finding a New Role

Lee had his urethra removed and a suprapubic catheter placed two years after surgery because of a recurrence of cancer in the urethra. One year after surgery, he increased his work time to eight hours, and returned to working, exercising and enjoying life as he had before the surgery. He believed that the surgery had completely cured him, so he began to drink again. One year after beginning to drink alcohol, he had a cancer relapse. Believing that alcohol was the cause of the relapse, he regretted drinking. After placement of the suprapubic catheter, he could no longer go to a public sauna or engage in his favourite sports, such as hiking and soccer. However, instead of playing soccer as a player, he became a coach and began expanding his role in the family that he had previously neglected. These new roles prompted Lee to perceive himself as a part of the society and a family member, rather than a patient. Lee was able to fulfil these roles through his own attempts, as well as the willingness of the people around him to gladly assign new roles to him.
Brother, you’re sick. You’re sick, so stay put. They don’t treat me like that. If people treat me like a patient, it discourages me. My wife- she also asked me, ‘honey, can you do some cleaning today?’ or ‘I couldn’t fold the laundry today’. I feel good because she asks me to do things. (omitted) If she told me ‘honey, you’re sick so just think about taking care of yourself and don’t do (anything)’, I may have been very depressed. I think it wouldn’t be good if I keep thinking ‘I’m a patient’, ‘I’m a patient’.(Interview 2, Lee)

Although living with a urine bag after relapse is physically uncomfortable, Lee liked his new life better and felt rewarded because the new life was good for everyone in the family, unlike his previous self-centred life.

#### 3.2.3. Park’s Story

Changes in Daily Life: Beginning of a Restrained Life

Park was a 58-year-old man. He was diagnosed with NMIBC and underwent TURBT 6 years prior to the study. He used to run a restaurant with his wife, but he quit following his wife’s suggestion, after the surgery. At the time, his wife and family implored him to quit smoking, but he was unable to give it up, and quitting work and staying home all day made him feel bored and hollow. He had to undergo 11 more rounds of TURBT because of continued cancer relapse and progression. One year and nine months before this study, he underwent cystectomy and neobladder reconstruction because of MIBC. After cystectomy, his wife reproached him, calling him “worse than the captain of the Sewol ferry who left kids behind and ran away” for lying to her and continuing to smoke. He was pushed to the brink of divorce, and finally reflected on and acknowledged his faults. He eventually quit smoking and endeavoured to remain considerate of his wife.

Adjustment Experiences: Living in Line with the Health Condition and Family Wishes

Park accepted and adjusted to the changes in his body and daily living. His incontinence did not subside, even after some time. He strove to adapt to his new life by using a diaper, performing self-catheterisation as per his doctor’s advice, packing diapers when going outside, trying to ignore the discomfort that accompanies incontinence and wearing a modernised hanbok to conceal his use of diapers.
It (urine) just leaks too much. The doctor tells me to wait a little longer, but I was doubtful … but I have no option but to just live with it.(Interview 1, Park)

His approach to changes in his sex life was similar. Although he wished to resume his sex life, he thought that it may be difficult for his wife, so he calmly accepted the change.
Well, as she sees me... I’m wearing a diaper and it stinks. I’m not sexually attractive anymore. After changing my diaper, I think I can have a sex life like before, but I don’t think she would. If I were in her shoes, I wouldn’t be able to approach me, I would be hesitant. After surgery, my sexual sense has decreased and I feel less desire as well; I’m also getting old. I have accepted that there has been a change.(Interview 1, Park)

Although he did not seek counselling at the time because he was caught up with taking care of his health, after adjusting to the continued incontinence, he began to consider sex counselling.
When I visited another urologist, he said patients like me can use Viagra. I want to consult the doctor. (The reason that I haven’t seen a doctor for it is) I didn’t have a chance to think of it due to the need to focus on my recovery. I don’t think much about sex life because I’m getting older. If I were young then I would have tried more.(Interview 2, Park)

In the past, Park loved drinking and smoking, playing golf and hanging out with people. In contrast, his spouse considered secular joy as something to be avoided. She engaged in good deeds and lived a life of sacrifice and prayer. The two completely different people once stood in conflict, but after his surgery, Park came to believe that his wife was right and began adjusting to her way of life.
Now I listen to her really well. I only do things that she says to do and don’t do things that she says not to do.(Interview 1, Park)

Meaning of adjustment experience: Repenting past faults and accepting the present

Park stopped working after the NMIBC diagnosis, which left him bored and in need of small chores; thus, he took charge of housekeeping and sometimes helped out with the restaurant, which he enjoyed, but also found uncomfortable because of his incontinence. In that sense, Park believed that not working helped him adjust to the changes. Although incontinence restricted social and leisure activities, he enjoyed inline skating and skateboarding in place of playing golf. He also enjoyed good food and played pool with his friends instead of drinking. He began to attend mass every day with his spouse, a devout Catholic, and committed to a life of faith. Despite discovering new activities to replace those he could no longer do, maintaining social relationships and continuing most aspects of life he had before the surgery, he felt that he had a poor quality of life. To Park, adjustment meant repenting his past faults, living in line with his health condition and family wishes and living selflessly, but he had not yet regained his family’s trust. Nevertheless, Park wished to continue with his current life, being considerate of others and understanding others with a big heart.
There’s nothing I can do. My wife has put me on her watch list. She controls where I should go and by what time I should come back home.(Interview 1, Park)
(My current life is) Not bad. I try to live within what is given. I try to adjust. If I just give up my thoughts a little bit, that comes around as a good thing for me, and to the people around me, so I’m okay now. It’s a little uncomfortable but I need to live with it because it is the fruit of my doing.(Interview 1, Park)

## 4. Discussion

This study examined the physical and mental changes, coping experiences and the consequent life changes and adjustment experiences among three Korean male bladder cancer survivors with a neobladder. The purpose was to obtain an in-depth understanding of the underlying meanings attached to the experiences and adjustments in this population.

Following neobladder reconstruction, all three participants had to urinate while seated and needed to strain and press on their abdomen. They also experienced lower urinary tract symptoms, such as residual urine, urinary frequency, nocturia and incontinence. Further, they experienced stress, including anxiety and frustration as a result of their incontinence, which is consistent with previous research findings [3,6]. Changes in sexual functioning were also common among the participants, which were also reported in previous studies [3,6]. Yet, a urinary tract infection (UTI) or stone formation—two frequently reported late complications after neobladder construction—were not reported in this study. Given that they are common complications after neobladder reconstruction [15,16], it is worth investigating the reasons for their non-occurrence further.

Like the patients in a previous study [3], the participants in this study coped with incontinence and the altered voiding mechanism with a variety of methods, such as using a diaper, doing Kegel exercises and performing self-catheterisation. In line with previous reports that people use different strategies to deal with incontinence and find their own methods because instructions for incontinence management are lacking [6], the participants in this study also developed solutions for themselves. This indicates that patients are not given adequate education and guidance on incontinence. Given that the focus during the patient’s hospital stay is on managing various complications that can arise immediately after surgery and that incontinence occurs as patients initiate self-urination at about the time of discharge, it is possible that the health care providers might not regard incontinence seriously. However, according to Ahmadi and his colleagues [17] only about 60% of male patients with a neobladder achieve daytime continence and about 45% achieve night-time continence one year after surgery. In other words, about half of bladder cancer survivors with a neobladder continue to live with incontinence. Therefore, health care providers need to offer more specific incontinence education and guidance to these patients.

Whereas they actively sought a variety of coping strategies for incontinence, the participants in this study gave up on recovery of sexual function without actively seeking a resolution. This finding differs from the results reported by Beitz and Zuzelo [3], where patients attempted to resolve sexual dysfunction through drug therapy or other means. Either the couples did not have an active sexual life even before surgery or the participants’ spouses were fully aware of the possibility of sexual problems through preoperative education. Therefore, they had little desire for a sex life, and it seems that the participants in this study gave up without finding a solution to sexual dysfunction. Middle-aged Koreans regard sex as an important part of their lives but very few individuals seek medical help when they have sexual problems [18]. This might be also related to the social norms in Korea where talking openly about sex and sexual issues with others is generally considered indecent. As a result, instead of pursuing sexual rehabilitation, the participants might have chosen not to talk about it as a face-saving attempt.

Further, adapting to the changed voiding mechanism is the priority immediately after surgery, so both patients and healthcare providers alike tend to put sex-related issues aside. A considerable amount of time has already passed by the time patients adapt to the altered voiding mechanism and later, survivors of bladder cancer may find it difficult to discuss sexual problems with healthcare providers and spouse. Thus, sex-related education and consultation must be provided continuously from the preoperative to postoperative adjustment period and should be designed to involve the patient’s spouse.

In addition to incontinence and reduced sexual function, the participants also experienced life changes. Stopping work is considered the greatest life change. One’s job and economic power strongly influence one’s identity as the head of the household among middle-aged Korean men [13]. While feeling burdened and stressed due to their financial responsibility toward the household, Korean men also feel as if their value is lost when they are not economically active [13]. Going by that, it was not surprising that the participants in this study wanted to keep working, implying that their desire could be the motivation for their effort to recover. The participant who did not work also acknowledged the need for work. In contrast, women cancer patients are responsible for childrearing and caring for their family, which is a burden, but it also drives their recovery and acceptance [19,20]. This is speculated to be a result of the traditional perception of gender roles among middle-aged and older adults [21]. These results suggest that male cancer survivors should consider continuing work to a certain degree, depending on their state of illness and work conditions, rather than quitting work without question.

At the same time, however, returning to work is difficult from a practical perspective for bladder cancer survivors with a neobladder. They need time to recover after surgery and to adjust to their changed voiding mechanism, but there is no regulation on paid sick leave in the Labor Standards Act. Hence, workers are allowed sick leave according to the organisational agreement or work rules at their workplace, and those who are not allowed sick leave must accept the possibility that they may lose their job when they undergo treatment [22]. In fact, 47% of cancer patients lose their jobs and the probability of reemployment is low in individuals aged 50 or older [23]. Negative perceptions and attitudes toward cancer survivors’ return to work [24] also hinders their ability to return. Thus, institutional support and a change in social perception are needed to assist cancer survivors to return to work after treatment. For bladder cancer survivors with a neobladder to return to work, accessible restrooms and health facilities, such as an on-site health clinic, are needed to allow them to change their diaper or clothes and perform self-catheterisation as necessary. Moreover, a flexible work environment that enables a phased return to work is needed at all workplaces. A past study on breast cancer survivors returning to work also suggested the need for a receptive work climate and social policies [25].

Another noticeable finding was family conflict associated with the participants’ continued practice of unhealthy behaviour. In this study, Kim, one the participants, continued to smoke and drink because he believed that stress was the major cause of his bladder cancer, not smoking or drinking. Kim’s unhealthy behaviour caused family reprimand. This result is similar to a prior finding that people who continue to smoke after surgery experience anger and concern from the people around them [3]. Thus, it is necessary to provide accurate information so that patients’ perception about the cancer aetiology does not have an adverse impact on their treatment and recovery. Cancer patients who smoke experience social stigma, and those with a disease that is considered to be caused by smoking, experience a particularly strong stigma [26]. Social stigma adversely affects cancer survivors’ mental health, causing depression and anxiety, and may contribute to smoking continuation [26]. Hence, families and healthcare providers should adopt an empathetic attitude about the difficulties of quitting smoking, as opposed to a stigmatising attitude. Interventions to stop unhealthy behaviours are also needed. Practical smoking cessation programmes that combine behavioural and pharmacological therapies should be provided, rather than simply recommending that patients quit smoking [26].

Experiences with adjusting to various changes after surgery differed across participants. Emerged themes included ‘actively explore resolutions’, ‘accept change and partial return to daily life’, and ‘living in line with health conditions and family wishes’. Cancer patients’ adjustment is influenced by an array of disease-related factors, such as location, type, stage and treatment, as well as personal and social factors, including character traits, coping strategies, emotional maturity, family, friends and community [8]. Although the participants had different adjustment experiences, one common theme was that all participants continued to engage in social relationships and social and leisure activities. To do so, they had to first disclose their difficulties to their close friends. Because lower urinary tract symptoms, such as urinary frequency, urgency and incontinence, may lead to stigma associated with frequent use of the restroom, smell and clothes stains, patients who have these symptoms sometimes choose secrecy and silence [27]. Silence as a defence against stigma may be a protective mechanism but is also an obstacle to seeking help [27]. In the current study, participants chose to open up about their incontinence to their close friends instead of remaining silent, which enabled them to maintain social relationships and continue to engage in social activities.

The meanings underlying the adjustment experiences of the participants in this study were ‘return to a daily life’, ‘new normal’ and ‘repenting past faults and accepting the present’. Kim returned to work and eventually expanded his career abroad, while other participants, Lee and Park, lived a changed life as family-oriented men. Cancer survivors can achieve additional success by reviewing their cancer experience, accepting the consequences, mourning the changes to their anticipated life and moving on with their adjusted life plans [28]. In previous studies, cancer patients came to value the days they would live as they looked back on the lives they had lived, and instead of living only for their themselves and their families, they sought out and approached others who needed their help [20,29]. Further, they experienced growth such as being grateful for life, changing their priorities, regaining confidence from overcoming pain and making their new life their own [19].

The overarching theme of the results of this study—where patients adopt a new attitude toward life and grow as they overcome various difficulties after surgery—is similar to that of previous studies. However, the adjustment experiences observed in this study are distinguished in that they reflect the characteristic narrative of middle-aged men, where the previous emphasis on work and social life was transformed to a family-oriented life or further bolstered as a work-oriented life. This suggests that cancer survivors’ social, cultural and institutional narratives must be considered when assisting in their adjustment and rehabilitation.

This study is noteworthy in that it sheds light on the adjustment experiences of male cancer survivors where there is a scarcity of research on male cancer survivors compared to the volume of data on female cancer survivors, such as breast and gynaecologic cancer survivors. In addition, by adding Korean male cancer survivors’ experience to the field, this study is expected to provide culture-enriched information to a certain extent. One limitation of this study is that it primarily deals with the positive adjustment experiences of male bladder cancer survivors with a neobladder. To broaden the understanding of this group, a future study can explore possible negative adjustment experiences among them. Similarly, a future study can study patients who were socially isolated because of incontinence or who had relationship difficulties with their spouses related to reduced sexual functioning. Finally, an in-depth exploration of female bladder cancer survivors after radical cystectomy is worth examining to provide gender-balanced information in regard to the topic.

## 5. Conclusions

This is the first study to examine the adjustment experiences of Korean male bladder cancer survivors with a neobladder. The findings have important implications for life changes and consequent adjustments beyond the physical postoperative symptoms. Further, we explored not only their personal experiences but also the social, cultural and institutional narratives underlying their experiences using narrative inquiry methodology. Bladder cancer survivors with a neobladder continue to adjust to changes in the voiding mechanism and various symptoms including incontinence even after surgical cancer resection. Therefore, we concluded that in addition to therapeutic interventions such as surgery, and chemotherapy, interventions are needed to assist with adjustment, rehabilitation and a return to society.

The active and open coping strategies demonstrated by this study’s bladder cancer survivors with a neobladder provide a reference for bladder cancer patients who are scheduled to undergo neobladder reconstruction. Support from family and friends by way of them adopting an empathetic attitude, rather than viewing them as people who developed cancer from smoking, is essential for promoting adjustment in these patients. In addition, it is crucial for nurses to provide education and interventions about health problems that accompany neobladder reconstruction, such as prolonged incontinence. Finally, institutional support and improved social awareness are needed to help bladder cancer survivors with a neobladder return to work and society following surgery.

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
