# Peer review of "A Narrative Inquiry into the Adjustment Experiences of Male Bladder Cancer Survivors with a Neobladder"

_ijerph, 2020, doi:10.3390/ijerph17218260_

Round 1

Reviewer 1 Report

Aim of the study could be interesting, considering minor attention paid in QoL changes than perioperative and oncologic outcomes. However, it is not sufficient evaluate personal opinions of only three patients underwent neobladder for MIBC. There isn't any objective evaluation made by self-assessed questionnaires that provides evidence based data on HRQoL. Moreover, there is an oncologic concern about the patient who underwent  just urethrectomy for cancer relapse, without any undiversion but just had a suprapubic catheter placed.

Reviewer 2 Report

This is a well written paper and very interesting to read. However, it is rather consistent with a case report and review in its present form. There are interesting hypothesis regarding the impact of lifestlye choices (e.g. smoking), however it appears difficult to support the conclusions based on the low patient number (n = 3) and purely descriptive study design.

Beitz et al. (Ref. 3) have published a similar study with 22 participants (13 men, 9 women) in Can Oncol Nurs J including patients with ileal conduit (5 men, 4 women) and neobladder (9 men, 4 women). If the authors intend to publish their results as a full paper, then the patient number must be increased. The narrative inquiry could be finalized by a quantitative evaluation (e.g. standardized questionaire) to support the conclusions.

The cases focus on the positive coping experience of male survivors with a neobladder. However, including gender aspects and negative coping would be essential for a balanced account in a research article. 

Minor comment:

Affiliation of Eunju Kim is missing on the title page.

P 2, L52-54 Four colons ":" in line?       

Reviewer 3 Report

This study aimed at exploring adjustment experiences after neobladder reconstruction in three male patients undergone radical cystectomy for bladder cancer. A narrative inquiry method was used to investigate the individual experiences of these men and the meaning of their experiences. Psychological and sociological post-operative changes caused by functional outcomes of radical cystectomy with neobladder are explored.

On the whole, the article is well written and results somehow interesting however the following consideration must be done:

1) the "scientificity" of the methods is questionable. Patients, in fact, were not administered validated questionnaires while their post-operative quality of life was assessed by means of a narrative inquiry which requires interpretation by the examiner. However, in the last four decades, we have witnessed an increasing use of validated, bladder cancer-specific questionnaires with UD-specific constructs (Rangarajan K, et al. Trends in quality of life reporting for radical cystectomy and urinary diversion over the last four decades: A systematic review of the literature. Arab J Urol 2019 Apr 14;17(3):181-194. doi: 10.1080/2090598X.2019.1600279. PMID: 31489233)

2) it is not possible to generalize findings obtained interviewing three men undergone radical cystectomy with orthotropic neobladder as the study population is extremely limited. Moreover, no info regarding their preoperative mental health status is available though it was found associated with the risk of post-operative complications (Pranav Sharma, Carl H. Henriksen, Kamran Zargar-Shoshtari, Ren Xin, Michael A. Poch, Julio M. Pow-Sang, Wade J. Sexton, Philippe E. Spiess, Scott M. Gilbert. Preoperative Patient Reported Mental Health is Associated with High Grade Complications after Radical Cystectomy. J Urol. 2016 Jan; 195(1): 47–52. doi: 10.1016/j.juro.2015.07.095 PMCID: PMC4924593

3) sexual behavior is known to be strictly related to partner's desire and attitude. Bearing this in mind, also the partners' reactions to the husbands' post-operative physical/psychological conditions deserve investigation.

Please have a look at:

  • Agustina Bessa, Rebecca Martin, Christel Häggström, Deborah Enting, Suzanne Amery, Muhammad Shamim Khan, Fidelma Cahill, Harriet Wylie, Samantha Broadhead, Kathryn Chatterton, Sachin Malde, Rajesh Nair, Ramesh Thurairaja, Pardeep Kumar, Anna Haire, Saran Green, Margaret Northover, Karen Briggs, Mieke Van Hemelrijck. Unmet needs in sexual health in bladder cancer patients: a systematic review of the evidence. BMC Urol. 2020; 20: 64. doi: 10.1186/s12894-020-00634-1. PMCID: PMC7268732
  • Maria Angela Cerruto, Carolina D’Elia, Giovanni Cacciamani, Davide De Marchi, Salvatore Siracusano, Massimo Iafrate, Mauro Niero, Cristina Lonardi, Pierfrancesco Bassi, Emanuele Belgrano, Ciro Imbimbo, Marco Racioppi, Renato Talamini, Stefano Ciciliato, Laura Toffoli, Michele Rizzo, Francesco Visalli, Paolo Verze, Walter Artibani. Behavioural profile and human adaptation of survivors after radical cystectomy and ileal conduit. Health Qual Life Outcomes. 2014; 12: 46. doi: 10.1186/1477-7525-12-46. PMCID: PMC3991923

4) it is unclear how the three men were selected. Apparently, all the enrolled patients suffered from a certain degree of urinary incontinence and erectile dysfunction. However, although a non-negligible share of men experience post-operative incontinence and impotence, these undesirable outcomes are reported by a minority of the treated subjects.

Please discuss outcomes of RARC with orthotropic neobladder as reported by Brassetti A, Anceschi U, Bertolo R, Ferriero M, Tuderti G, Capitanio U, Larcher A, Garisto J, Antonelli A, Mottire A, Minervini A, Dell'oglio P, Veccia A, Amparore D, Flammia RS, Mari A, Porpiglia F, Montorsi F, Kaouk J, Autorino R, Carini M, Gallucci M, Simone G. Surgical quality, cancer control and functional preservation: introducing a novel trifecta for robot-assisted partial nephrectomy. Minerva Urol Nefrol. 2020 Feb;72(1):82-90. doi: 10.23736/S0393-2249.19.03570-7. Epub 2019 Dec 12. PMID: 31833720.

5) this study only investigates functional outcomes of radical cystectomy with neobladder urinary diversion. However, only a limited share of patients undergoing radical cystectomy for bladder cancer are offered continent diversion, also in tertiary referral centers.

Please discuss and cite: Brassetti A, Möller A, Laurin O, Höijer J, Adding C, Miyakawa A, Hosseini A, Wiklund P. Evolution of cystectomy care over an 11-year period in a high-volume tertiary referral centre. BJU Int. 2018 May;121(5):752-757. doi: 10.1111/bju.14112. Epub 2018 Jan 19. PMID: 29281852.

Round 2

Reviewer 1 Report

The manuscript was notably improved, however few further modifications are needed before the final acceptance.

  • The presence of residual urine after catheterization or urinary retention for incomplete voiding are risk factors for lithiasis and infection. Did any of the three patients experience UTI or stone formations? Even if not, these are notable events after an orthotopic diversion having a significant impact on daily life. This topic should be discussed (consider references Ferriero et al 2014).
  • Sexual health issue after radical cystectomy in male patients has been widely addressed with seminal and nerve sparing techniques; however even in female patients, where sexual function preservation was previously forclosed, sex sparing technique was introduced as alternative to standard radical cystectomy, when oncologically feasible and strong motivation occurred. (see Tuderti et al 2020)

Reviewer 2 Report

The authors argue, that sample size is not relevant for a narrative inquiry and their purpose was not to prove hypotheses but to understand experiences.

Still, the present outline appears biased towars single experiences and not inclusive with respect to gender. There are essentially no changes, that have been made to the manuscript with respect to the criticism.     

Reviewer 3 Report

The "scientific significance" of the narrative inquiry method still remains difficult to understand for me because results cannot be generalized.

However, also the significance of this comment is limited as the reviewer is not a psychologist nore a sociologist but only a urologist, whose area of expertise is limited to urologic Oncology. 

Personal limitations of the present reviewer, however, should not influence its judgment. The manuscript, at the end, is well written and discussion is interesting. 

I am heartily convinced that such new research methods should be further investigated and deserve being published on impacted journals to gain popularity. 
